# The Relationship between Maternal Antibodies to Fetal Brain and Prenatal Stress Exposure in Autism Spectrum Disorder

**DOI:** 10.3390/metabo13050663

**Published:** 2023-05-16

**Authors:** Amy N. Costa, Bradley J. Ferguson, Emily Hawkins, Adriana Coman, Joseph Schauer, Alex Ramirez-Celis, Patrick M. Hecht, Danielle Bruce, Michael Tilley, Zohreh Talebizadeh, Judy Van de Water, David Q. Beversdorf

**Affiliations:** 1Department of Psychological Sciences, University of Missouri, Columbia, MO 65211, USA; ancosta@mail.missouri.edu (A.N.C.); ekhb97@mail.missouri.edu (E.H.); 2Department of Health Psychology, University of Missouri, Columbia, MO 65211, USA; fergusonbj@health.missouri.edu; 3Thompson Center for Autism and Neurodevelopment, University of Missouri, Columbia, MO 65211, USA; 4Interdiscipinary Neuroscience Program, University of Missouri, Columbia, MO 65211, USA; patrick.m.hecht@gmail.com; 5Department of Biological Sciences, University of Missouri, Columbia, MO 65211, USA; 6Department of Biochemistry, Grinnell College, Grinnell, IA 50112, USA; comanadr@grinnell.edu; 7Department of Internal Medicine, Division of Rheumatology, Allergy, and Immunology, University of California, Davis, CA 95161, USA; jdschauer@ucdavis.edu (J.S.); aleramirezcelis@ucdavis.edu (A.R.-C.); javandewater@ucdavis.edu (J.V.d.W.); 8Department of Biology, Central Methodist University, Fayette, MO 65248, USA; dbruce@centralmethodist.edu (D.B.); mtilley@minutemanpress.com (M.T.); 9The American College of Medical Genetics and Genomics, Bethesda, MD 20814, USA; ztalebizadeh@acmg.net; 10Departments of Radiology and Neurology, University of Missouri, Columbia, MO 65212, USA

**Keywords:** autism spectrum disorder, gene expression, stress, immune system, 5-HTTLPR

## Abstract

Environmental and genetic factors contribute to the etiology of autism spectrum disorder (ASD), but their interaction is less well understood. Mothers that are genetically more stress-susceptible have been found to be at increased risk of having a child with ASD after exposure to stress during pregnancy. Additionally, the presence of maternal antibodies for the fetal brain is associated with a diagnosis of ASD in children. However, the relationship between prenatal stress exposure and maternal antibodies in the mothers of children diagnosed with ASD has not yet been addressed. This exploratory study examined the association of maternal antibody response with prenatal stress and a diagnosis of ASD in children. Blood samples from 53 mothers with at least one child diagnosed with ASD were examined by ELISA. Maternal antibody presence, perceived stress levels during pregnancy (high or low), and maternal 5-HTTLPR polymorphisms were examined for their interrelationship in ASD. While high incidences of prenatal stress and maternal antibodies were found in the sample, they were not associated with each other (*p* = 0.709, Cramér’s *V* = 0.051). Furthermore, the results revealed no significant association between maternal antibody presence and the interaction between 5-HTTLPR genotype and stress (*p* = 0.729, Cramér’s *V* = 0.157). Prenatal stress was not found to be associated with the presence of maternal antibodies in the context of ASD, at least in this initial exploratory sample. Despite the known relationship between stress and changes in immune function, these results suggest that prenatal stress and immune dysregulation are independently associated with a diagnosis of ASD in this study population, rather than acting through a convergent mechanism. However, this would need to be confirmed in a larger sample.

## 1. Introduction

The developmental origins of health and disease (DOHaD) hypothesis proposes that the environment during in utero development impacts the health of offspring [1]. Recent studies have provided support for this theory, demonstrating that environmental factors during the prenatal period affect neurobiological developments, including those that are associated with autism spectrum disorder (ASD; [2]). Genetics is widely recognized as a critical factor in the development of ASD [3,4]. While more research has clarified the genetic factors associated with ASD, prenatal environmental factors and the interaction between genetic and environmental factors are less understood [5].

Exposure to prenatal stress in mothers during pregnancy impacts behavioral and developmental outcomes in their children. For example, schizophrenia and emotional disturbances have been associated with maternal stress [6,7]. Recent evidence also supports the theory that maternal stress exposure is a factor in the development of ASD [8,9]. Larger epidemiological studies also support the relationship between prenatal stress and ASD [10,11]. Although a large Danish national registry study reported no association between maternal bereavement and ASD [12], an association was observed between maternal bereavement and ASD in this study prior to accounting for covariates such as maternal psychiatric conditions [12]. Other reports examining data from a Danish national registry study found that maternal psychiatric conditions were one of the strongest prenatal risk factors for ASD [13]. Reports examining data from a Swedish registry study have also revealed a relationship between 3rd-trimester maternal stress exposure and the risk of ASD [14]. Furthermore, reports examining data from the Nurses’ Health Study revealed that maternal exposure to partner abuse during pregnancy is strongly associated with ASD, although the timing of exposure with the strongest association with ASD was found to be earlier in gestation [9]. Finally, a recent meta-analysis has supported an association between prenatal maternal stress and the risk of ASD [15], and a subsequent large-population study in China also confirmed an association between prenatal stress exposure and the development of autistic-like behaviors [16].

The timing of maternal psychosocial stressors associated with major life events that occur during pregnancy appears to be an important risk factor in the development of ASD. For example, one study found a higher overall incidence of stressors among mothers of children with ASD at the end of the second to the beginning of the third trimester [8]. A relationship between the incidence and severity of tropical storms, which serve as naturally occurring environmental stressors, and the incidence of the birth of children with ASD has also been shown, with the association found to be strongest at a similar point in gestation [17].

However, psychological stress during pregnancy and a child’s exposure to maternal stress in utero does not always lead to a diagnosis of ASD in the fetus. In fact, a significant portion of mothers who experience stressful events throughout pregnancy have neurotypical children. A gene × environment (G × E) interaction may explain why some mothers that undergo stressful events have children with ASD and some do not. The serotonin reuptake transporter gene (SLC6A4) is known to have a role in stress reactivity [11]. The SLC6A4 gene encodes the SERT protein and contains its associated promoter region 5-HTTLPR. SERT transports extracellular serotonin back into the neuron [18]. The promoter, 5-HTTLPR, has either a long (L) or a short (S) allele [18]. The presence of the S-allele has been linked to many psychological outcomes [19,20]. Serotonin transporter polymorphisms are associated with major depressive disorder, social phobia, and agoraphobia, and have also been associated with substance use disorders [21]. Subsequent research revealed that the presence of at least one copy of the S-allele may be a genetic risk factor for increased maternal stress response, leading to the development of ASD in children [22].

Stress can have a sizeable impact on immune function [23,24], and a perturbation in the maternal immune system during gestation, such as that resulting from a severe infection, has been thought to increase the risk of ASD in children [25,26]. Under normal conditions, the maternal immune system maintains a minimum of pathogens, while minimizing the inflammatory environment for the developing fetus [27,28]. Disruptions to gestational immune regulation, including the production of autoantibodies, can have adverse developmental effects on the fetus. Research has found an association between maternal fever and infection around the time of pregnancy and an increased risk of neurodevelopmental disorders, including ASD [26,29,30,31]. The diversity of maternal infections associated with neurodevelopmental disorders suggests that the maternal immune response may be a link between sickness in the mother and altered neurodevelopment in the child [10,32]. The production of cytokines is a key driver of the maternal immune response to pathogens responsible for the signaling immune and other cells to respond to infection. Cytokines are involved in a plethora of aspects of neurodevelopment, and some maternal cytokines can cross the placenta or act on placental cells to stimulate the production of immune mediators in the fetal compartment. Dramatic fluctuations in cytokine and chemokine levels can alter neurodevelopment, potentially resulting in an ASD diagnosis [10,33]. Specifically, it has been shown that elevation in mid-gestational levels of pro-inflammatory cytokines is associated with an increased risk of ASD diagnosis in children [34,35].

The maternal autoantibody response is also a factor thought to play a role in the development of ASD. In utero, children are supplied with maternal antibodies, which are essential for protecting the fetus [36]. Previous studies have hypothesized that the placental transfer of maternal antibodies could interfere with brain development and potentially lead to an increased risk of ASD [37,38]. Maternal autoantibodies reactive towards fetal brain proteins have been observed in nearly one-fourth of mothers of children with ASD, compared to less than 1% in mothers of unaffected children [39,40,41]. Specifically, lactate dehydrogenase A and B (LDH-A, LDH-B), collapsin response mediator proteins 1 and 2 (CRMP1, CRMP2), Y-box binding protein 1 (YBX1), stress-induced phosophoprotein 1 (STIP1), and guanine deaminase (GDA) have been identified as specific autoantibodies that are reactive towards the fetal brain and that are associated with an outcome of ASD in children (termed maternal autoantibody-related or MAR autism [42,43,44]).

While it is known that stress impacts immune function [23], to our knowledge, no studies have examined how prenatal stress, 5-HTLLPR genotypes, and their interaction are associated with the maternal immune response in the context of an ASD diagnosis. It is important to understand if environmental factors such as stress may interact with immune function and, in tandem with genetic factors, contribute to ASD risk. The present exploratory study examined whether an ASD-directed maternal autoantibody response was related to prenatal stress in their respective associations with ASD. Additionally, the study explored whether the interaction between prenatal stress and the presence of at least one copy of the S allele on 5-HTTLPR genotyping is associated with the maternal autoantibody response.

## 2. Methods

Families with a child diagnosed with ASD under the age of ten years old from the University of Missouri Thompson Center for Autism & Neurodevelopmental Disorders database were contacted. All participants with a child with ASD without a known genetic cause who were willing to participate were recruited. The children of the families were all below ten years of age (average age = 6.8 ± 1.8) to maximize the parents’ ability to recall information from the prenatal period. Families were invited to provide samples for genetic analysis and complete a questionnaire regarding the prenatal period. Previously collected blood samples from fifty-three mothers who had at least one child diagnosed with ASD were included in the study [22]. All ASD diagnoses were confirmed via Autism Diagnostic Interview-Revised (ADI-R, [45]) and/or Autism Diagnostic Observation Scale (ADOS, [46]) scores. The study was conducted according to the guidelines of the Declaration of Helsinki and approval was sought from the University of Missouri Health Sciences Review Board (Project # 1106357). Informed consent was obtained from all participants involved in the study. Data will be made available in a de-identified manner upon request. Raw data will not be publicly available for privacy reasons. The procedure for the current study was similar to that of previously cited work [22]. At the time of the appointment with the experimenter, the mothers completed questionnaires regarding their child with ASD and the gestational period of that child, which were derived from previous work [8]. Survey questions obtained information on the child’s birth date and the pregnancy length, in addition to the occurrence and subjective severity of major stressful events during or within one year of the pregnancy [22]. A list of common stressors [47] was provided to the subjects to facilitate the recall of events that may have occurred during the pregnancy. The stressors included psychosocial stressors such as divorce, death in the family, or loss of a job [22,47]. Blood was drawn via a standard venipuncture from the median cubital vein of the arm. If blood was unable to be drawn, buccal swabs were collected from the subject’s cheek. Genomic DNA was obtained from either subjects’ whole blood (Flexigene kit; Qiagen, Hilden, Germany) or cheek swabs (QIAamp kit; Qiagen, Hilden, Germany) according to the manufacturer’s instructions. PCR was performed as previously described [22]. Briefly, the promoter region of the serotonin transporter gene was amplified using the Qiagen PCR kit from 25 ng genomic DNA using the following primers: 5′-TCCTCCGCTTTGGCGCCTCTTCC-3′ (Forward) and 5′-TGGGGGTTGCAGGGGAGATCCTG-3′ to identify the L and S alleles of 5-HTTLPR. Cycling conditions were as follows: 95C for 15 min followed by 35 cycles of 94C for 30 s, 65.5C for 90 s, and 72C for 60 s, with a final extension step of 72C for 10 min. PCR products were then loaded onto a 3.5% agarose gel and run for 1 h at 160 V. Bands were visualized with SYBR-safe DNA gel staining (Invitrogen), with 469 bp and 512 bp identifying the short and long alleles, respectively, as described previously [22].

The MAR IgG antibody reactivity of plasma samples against each antigen was determined by ELISA (enzyme-linked immunosorbent assay) using commercially available proteins, and the assay conditions were optimized for each protein as previously described [44]. Briefly, the autoantibody reactivity of the plasma samples against protein antigens was determined by Enzyme-Linked Immunosorbent Assay (ELISA) and corroborated by Western Blot (WB) using commercially available proteins, as described previously [44]. The protein concentration and plasma sample dilutions were optimized for each antigen for both assays. Microplates were coated with 100 μL of antigen (1.5–3 µg/µL) in a carbonate coating buffer pH 9.6, incubated overnight at 4 °C, washed four times with Phosphate Buffered Saline Tween-20 (PBST) 0.05%, and blocked with 2% Super Block (Thermo Scientific, Rockford, IL, USA) for 1 h at RT. The plasma samples were then diluted 1:250–1:1000 (depending on which antigen was being tested) and run in duplicate. Following dilution, 100 µL of the diluted sample was then added to each well, incubated for 1.5 h, washed four times in PBST 0.05% and then four times with (PBST) 0.05%, and then incubated with goat anti-human IgG-HRP IgG (Kirkegaard & Perry Laboratories, Inc., Gaithersburg, MA, USA) diluted 1:10,000 for 1 h. The plates were then washed four times with (PBST) 0.05%, and detection was performed by adding 100 µL of BD optEIA liquid substrate for ELISA (BD Biosciences, San Jose, CA, USA). After 4 min, the reaction was then stopped with 50 µL of 2N HCl. The absorbance was measured at 490–450 nm using an iMark Microplate Absorbance Reader (Biorad, Hercules, CA, USA).

After plate-plate normalization, a positive cut-off was established for each antigen using a ROC curve and Youden’s index as previously described. The positive control samples used to create the ROC were not included in the analysis.

## 3. Statistical Analyses

Statistical analyses were conducted to determine whether prenatal stress was associated with the presence of maternal autoantibodies and whether the interaction of prenatal stress and the 5-HTTLPR genotype was associated with maternal autoantibody presence via a Chi-squared test for independence. In the case of a significant interaction, simple slopes were examined to further understand the effect (i.e., S/S vs. S/L genotype). For prenatal stress, groups were broken into high (3+ stressors) and low (2 > stressors) prenatal stress. This approach to categorizing stress exposure has been successful in previous work demonstrating the miRNA profiles of prenatal stress exposure in ASD [48]. For the presence of maternal autoantibodies, groups were divided into the presence or absence of specific sets of antibodies previously found to be associated with ASD in approximately 20% of cases [11]. Additionally, groups were divided based on 5-HTTLPR genotypes and prenatal stress levels, as shown in Table 1.

## 4. Results

### Participant Characteristics

The average number of stressors reported was 2.03. There were 24 mothers who experienced 3 or more stressors, of whom 17 had at least one copy of the S allele. There were 11 mothers who were MAR-positive (Table 1 and Table 2). The specific ASD-associated autoantibody patterns [42,43,44] identified were as follows:

Three participants had the ASD-associated neuron-specific enolase (NSE) + STIP1 pattern;

Two had the CRMP2 + GDA pattern;

One had the CRMP1 + STIP1 pattern;

One had the CRMP1 + GDA pattern;

One had the CRMP1 + YBX1 pattern;

Three had multiple ASD-associated patterns (one with the CRMP2 + GDA and CRMP2 + STIP1 patterns, one with the CRMP1 + CRMP2, CRMP1 + GDA, and CRMP2 + GDA patterns, and one with the CRMP2 + GDA, CRMP2 + STIP1, GDA + YBOX1, STIP1 + NSE, LDHA + YBOX1, and LDHB + YBOX1 patterns).

The distribution of the number of reported stressors, maternal genotypes, and the presence of any maternal autoantibody patterns associated with ASD indicated that the association between the presence of maternal autoantibodies and prenatal stress groups was not significant (*p* = 0.709, Cramér’s *V* = 0.051, see Figure 1a). Furthermore, the results revealed no association between the presence of MAR and stress groups, and the association between the 5-HTTLPR genotype and stress groups was also not significant (*p* = 0.729, Cramér’s *V* = 0.157, see Figure 1b).

## 5. Discussion

Considering that stress affects immunity and the antibody response [23,49] and that MAR autoantibodies show an increased ASD risk [42], the present study examined whether prenatal stress was associated with the maternal autoantibody response. While the incidence of MAR autoantibody patterns (see above) and prenatal stress in the presence of the S-allele was high, the results revealed that prenatal stress was not associated with the presence of MAR autism, at least in this exploratory sample. Additionally, the interaction between the 5-HTTLPR polymorphism and prenatal stress was not associated with the presence of maternal autoantibodies in their association with ASD in this exploratory sample. Therefore, despite the interrelationships between prenatal stress, 5-HTTLPR polymorphism, and immune dysregulation [10], these factors do not appear to be related in their association with ASD in this exploratory sample.

These results begin to suggest that prenatal stress and maternal autoantibodies to fetal brain proteins are independently related to ASD with independent pathogenic pathways. As described above, previous studies have found prenatal stress to affect behavioral and neurodevelopmental outcomes in humans, including ASD diagnosis [11]. Additionally, past research has revealed that susceptibility to stress (via SLC6A4, specifically the 5-HTTLPR region) interacts with prenatal stress, shows an increased risk of ASD [22], and is associated with a distinct miRNA profile [48]. Other G × E interactions with stress have been observed in a range of neuropsychiatric conditions [5]. With respect to the ASD-specific maternal autoantibodies associated with an increased risk of ASD [42], recent evidence has revealed important information regarding their effects on regional brain volume and metabolites [50], and salient maternal antibodies to the fetal brain have recently been identified in samples from the Simons Simplex Collection [51], the association between maternal autoantibodies to the fetal brain and prenatal stress exposure is not known. We hypothesized that prenatal stress and maternal antibody response might be related due to the known relationship between stress and immune function [23]. Our findings suggest that prenatal stress, and its interaction with the 5-HTTLPR genotype, appears to be independent of maternal autoantibody response with respect to an ASD diagnosis. It is possible that the ASD-related maternal autoantibodies are in place prior to prenatal stress exposures and, therefore, the subsequent stress would not have the same impact on the formation of these autoantibodies.

These findings may serve as preliminary evidence that prenatal stress and maternal immune dysregulation are independently related to ASD diagnoses in offspring, but further investigation is warranted. Given that the present study was cross-sectional, future studies should monitor mothers during pregnancy to avoid the retrospective recall of stressors, better examine other aspects of maternal immune dysfunction, and understand the causal relationship between these biological events. Additionally, future studies should investigate the different genotypes associated with stress reactivity and maternal autoantibodies to determine if the interaction between prenatal stress and maternal autoantibodies is different for different stress-associated genes. While the sample size was small, previous studies of a similar size revealed miRNA results associated with prenatal stress in ASD [48]. However, the small sample size limits the conclusions that can be drawn from these findings. Future studies with larger samples would need to be performed to confirm this, and such larger studies could also explore whether there might be an association between prenatal stress and individual, specific MAR patterns. Additionally, larger studies would need to sample a more diverse population, as the distribution of the polymorphisms varies considerably across racial and ethnic groups [52].

The current study suggests that prenatal stress and the 5-HTTLPR genotype may be independent of the presence of autoantibodies in their relationship to ASD. Although the sample size was small and follow-up longitudinal studies should be conducted, this finding is one step towards a better understanding of the interaction of the various risk factors associated with ASD. Future studies will need to examine how other factors interact in their impact on ASD. In addition to the previously discussed interactions between genes and the environment, particularly for prenatal stress and neurodevelopment [5,22], recent evidence has shown that prenatal stress exposure and environmental exposure to air pollution interact in their impact on neurodevelopment [53]. Several other environmental factors that may occur during the prenatal period are associated with an increased incidence of ASD; this could be explored in other studies, in addition to genetic factors. Maternal exposure to pollutants, as mentioned above, has consistently been associated with ASD specifically for maternal exposure to air pollutants [54,55,56,57], with additional evidence of an interaction between air pollutant exposure and polymorphisms of the tyrosine kinase MET receptor gene [54]. An increased risk of ASD has also been associated with exposure to medication use in pregnancy, most notably for valproic acid [58]. Previous research has also reported an increased risk of ASD in association with prenatal exposure to β2-adrenergic agonists, commonly used to arrest premature labor, with an interaction between drug exposure and maternal polymorphisms in the β2-adrenergic receptor [59]. Other risk factors are being explored and identified including pesticides, endocrine-disrupting chemicals such as phthalates and bisphenol A, and maternal dietary factors, including a lack of folate supplementation during early pregnancy [60,61,62]. Increased parental age and short intervals between pregnancies have also been associated with an increased risk of ASD [63,64]. The interaction between individual risk factors will be important for a better understanding of the epidemiology of ASD. Understanding these interactions may also facilitate the determination of common and orthogonal mechanistic pathways, which can, hopefully, eventually lead to meaningful clustering of individuals that might respond to treatments targeting a common mechanism for a precision medicine approach for optimizing the treatment of ASD. However, we must acknowledge the complicated nature of the exploration of environmental factors, such as disentangling the effect of the medical condition necessitating the administration of valproic acid in the relationship between maternal valproic acid exposure and ASD, in addition to similarly potential confounding factors for other medication, dietary, and putative risk factors. The findings described herein suggesting that prenatal stress exposure and the presence of maternal autoantibodies are independent could be of some importance, though, as each factor appears to be present in a substantial proportion of ASD cases. Therefore, in aggregate, these factors may account for a significant proportion of ASD. However, future studies will need to include control cases without ASD in order to meaningfully determine the proportion of ASD associated with these factors in aggregate. Additionally, further work will be critical to better understand the downstream metabolic pathways mediating the effects of these factors impacting neurodevelopment, allowing for the potential development of meaningful interventions.

## 6. Conclusions

Environmental and genetic factors have been shown to contribute to the development of ASD, but how they might be interrelated is less well understood. Our results suggest that prenatal stress and immune dysregulation, while independently associated with ASD, are not interrelated in their association with ASD, at least based on this exploratory sample. This suggests that prenatal stress and immune dysregulation are independently associated with ASD, rather than acting through a common mechanism.

## Figures and Tables

**Figure 1 metabolites-13-00663-f001:**
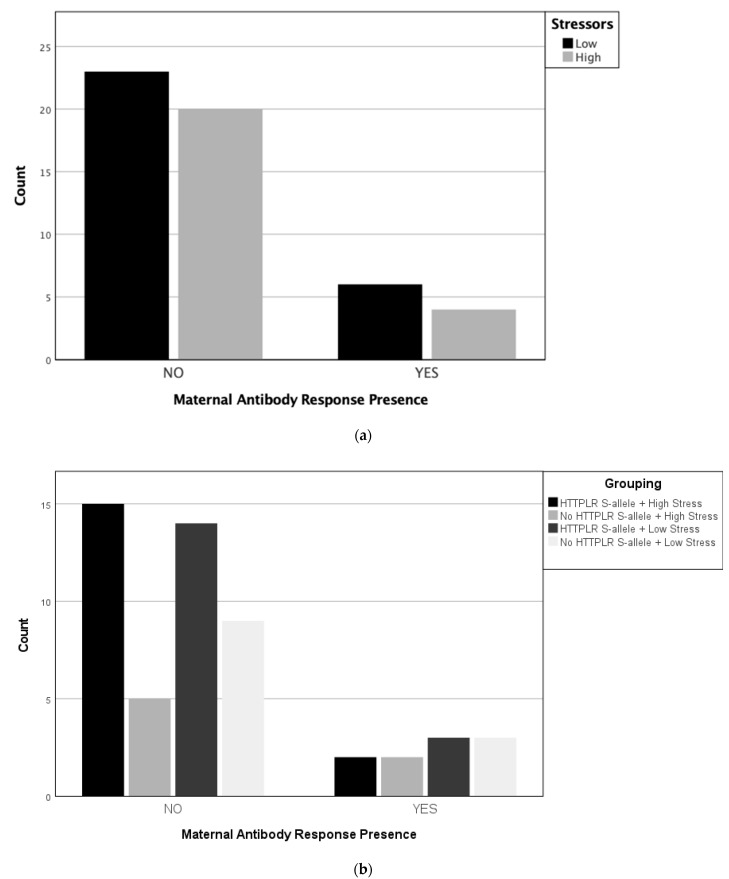
(**a**). Number of mothers with high and low prenatal stress exposure among mothers with or without maternal autoantibodies (**b**). Number of mothers with high and low prenatal stress exposure and the presence and absence of the HTTPLR S-allele among mothers with or without maternal autoantibodies.

**Table 1 metabolites-13-00663-t001:** Comparison Groups.

Prenatal Stress	5-HTTLPR Genotype	N
High	S/S + S/L	17
	LONG	7
Low	S/S + S/L	17
	LONG	12

Samples were divided into 4 groups based on 5-HTTPLPR genotype presence and prenatal stress levels.

**Table 2 metabolites-13-00663-t002:** Characteristics of study populations. Average ± standard deviation.

	*N* = 53
Maternal Age (years)	34.8 ± 6.4
Age of Child (years)	6.8 ± 1.8
Male gender (%)	89.8%
Maternal 5-HTTLPR genotype:	
S/S	12 (22.6%)
S/L	22 (41.5%)
L/L	19 (35.8%)
Number of Reported Stressors:	
Low (0–2)	29
High (3+)	24
Maternal autoantibody	11 (21%)

## Data Availability

Data will be made available in a de-identified manner upon request. Raw data will not be publicly available for privacy reasons.

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
