# Peer review of "The Relationship between Maternal Antibodies to Fetal Brain and Prenatal Stress Exposure in Autism Spectrum Disorder"

_metabolites, 2023, doi:10.3390/metabo13050663_

Round 1

Reviewer 1 Report

This is a paper that needs to be published.  It is well written, carefully researched and calls attention to the very substantial complexity of understanding predictors that might lead to the birth of a child with autism spectrum disorder.  

The only limitation I would note, and ask the authors to comment on, is the relatively small sample size.  These findings might or might not be robust to a larger sample, but as the study is carefully conducted, in my opinion it is well worth publishing even if the findings turn out not to be generalizable.

Author Response

We agree- we have now placed a greater emphasis on this implications of the modest sample size, and its implications for generalizability.

Reviewer 2 Report

Manuscript is well written and organized. I recommend acceptance after minor changes.

1. Page 2, "Prenatal stress impacts behavioral and developmental outcomes in their children" This sentence not clear. Whose children?

2. Please check whole manuscript in view of grammaticcal errors

Author Response

Thank you! We have now clarified this sentence and have reviewed for grammatical errors.

Reviewer 3 Report

Dear editor,

The study ''Relationship Between Maternal Antibodies to Fetal Brain and Prenatal Stress Exposure in Autism Spectrum Disorder'' is an interesting and well-written manuscript. Although the sample size is very small, the number of patients is insufficient for this conclusion given the prevalence of autism. In addition, it does not make a scientific contribution to Beeversdorf's previous articles on prenatal stressors, epigenetic factors, and autism.

Author Response

We understand this concern and have now placed a greater emphasis on this implications of the modest sample size, and its implications for generalizability.  We emphasize that this is an exploratory study, and discuss that it needs to be re-examined in a larger population.

Reviewer 4 Report

1.      INTRODUCTION:

Reduce de introduction, it is so long and the objective of paper is to correlation between ASD and 5-HTLLPR. In this section describe the relation between 5-HTLLPR polymorphism and SLC6A4 which is described associated to other mental disorders, and differences between LL/SS alleles in American and European-Americans populations, frequencies

2.      METHODS:

It is not included which genotype is studied. Two primers are described but corresponding to…?

In the statistical analysis authors described the determination of miRNA, lack of correlation with hypothesis 5-HTLLPR polymorphism and ASD.

Paragraph: For prenatal stress, groups were broken into high prenatal stress (3+ more stressors) and low pre-natal stress (2> stressors), as determined by previous work demonstrating miRNA profiles of prenatal stress exposure in ASD (Beversdorf et al., 2021). Difficult to understand

How were selected the patient?

Define the category stressors

3.      RESULTS

Paragraph: Among the ASD-associated autoantibody patterns, three had the neuron-specific enolase (NSE) + STIP1 pattern, two had CRMP2 + GDA, one had CRMP1 + STIP1, one had CRMP1 + GDA, one had CRMP1 + YBX1, and three had multiple ASD-associated patterns (one with the CRMP2 + GDA and CRMP2 + STIP1 pattern, one with the CRMP1 + CRMP2 and CRMP1 + GDA and CRMP2 + GDA patterns, and one with the CRMP2 + GDA and CRMP2 + STIP1 and GDA + YBOX1 and STIP1 + NSE and LDHA + YBOX1 and LDHB + YBOX1 patterns). Difficult to understand what are the correlation from these pattern?

4.      DISCUSSION

-was not associated with the presence of MAR autism, describe better MAR antibodies and autism

-maternal antibody response might be related due to the known relationship between stress and immune function (Kiecolt-Glaser et al., 2007). To our knowledge, no studies have examined the relationship between these two factors. It is described: 1/ Altered behavior, brain structure, and neurometabolites in a rat model of autism-specific maternal autoantibody exposure.Matthew R Bruce et al  Mol Psychiatry. 2023 Mar 27.doi: 10.1038/s41380-023-02020-3. 2/ Novel maternal autoantibodies in autism spectrum disorder: Implications for screening and diagnosis.Mazón-Cabrera R,et al. Front Neurosci. 2023 Feb 2;17:1067833. doi: 10.3389/fnins.2023.1067833. eCollection 2023. PMID: 36816132.

- Increased risk of ASD has also been associated with exposure to medication use in pregnancy most notably for valproic acid: are they taken into account to select patients? It is not clear in the methodology section. Also for the maternal dietary factors, many confusion factors/interactions..

Author Response

  1. INTRODUCTION:

Reduce de introduction, it is so long and the objective of paper is to correlation between ASD and 5-HTLLPR. In this section describe the relation between 5-HTLLPR polymorphism and SLC6A4 which is described associated to other mental disorders, and differences between LL/SS alleles in American and European-Americans populations, frequencies

Response: We were required to meet the word limit for a Research paper, so we needed to include an expanded Introduction.  We have now added the association with other mental disorders and the ethnic variation.  We added the ethnic variation to the discussion as this seems to be a better place for it.

  1. METHODS:

It is not included which genotype is studied. Two primers are described but corresponding to…?

In the statistical analysis authors described the determination of miRNA, lack of correlation with hypothesis 5-HTLLPR polymorphism and ASD.

Paragraph: For prenatal stress, groups were broken into high prenatal stress (3+ more stressors) and low pre-natal stress (2> stressors), as determined by previous work demonstrating miRNA profiles of prenatal stress exposure in ASD (Beversdorf et al., 2021). Difficult to understand

How were selected the patient?

Define the category stressors

Response: we have clarified the genotype, clarified the statistical analysis issue where the miRNA is mentioned, edited the problematic paragraph, and discuss patient recruitment, as well as the stressors.

  1. RESULTS

Paragraph: Among the ASD-associated autoantibody patterns, three had the neuron-specific enolase (NSE) + STIP1 pattern, two had CRMP2 + GDA, one had CRMP1 + STIP1, one had CRMP1 + GDA, one had CRMP1 + YBX1, and three had multiple ASD-associated patterns (one with the CRMP2 + GDA and CRMP2 + STIP1 pattern, one with the CRMP1 + CRMP2 and CRMP1 + GDA and CRMP2 + GDA patterns, and one with the CRMP2 + GDA and CRMP2 + STIP1 and GDA + YBOX1 and STIP1 + NSE and LDHA + YBOX1 and LDHB + YBOX1 patterns). Difficult to understand what are the correlation from these pattern?

Response: We agree that this is hard to follow.  We have re-structured for clarity.

  1. DISCUSSION

-was not associated with the presence of MAR autism, describe better MAR antibodies and autism

-maternal antibody response might be related due to the known relationship between stress and immune function (Kiecolt-Glaser et al., 2007). To our knowledge, no studies have examined the relationship between these two factors. It is described: 1/ Altered behavior, brain structure, and neurometabolites in a rat model of autism-specific maternal autoantibody exposure.Matthew R Bruce et al  Mol Psychiatry. 2023 Mar 27.doi: 10.1038/s41380-023-02020-3. 2/ Novel maternal autoantibodies in autism spectrum disorder: Implications for screening and diagnosis.Mazón-Cabrera R,et al. Front Neurosci. 2023 Feb 2;17:1067833. doi: 10.3389/fnins.2023.1067833. eCollection 2023. PMID: 36816132.

Increased risk of ASD has also been associated with exposure to medication use in pregnancy most notably for valproic acid: are they taken into account to select patients? It is not clear in the methodology section. Also for the maternal dietary factors, many confusion factors/interactions..

Response: we have clarified the MAR antibody association in the Discussion, clarified what we intended with the stress and immune function- stress and immunity are clearly related, the maternal antibody association was merely the hypotheses we were testing- this is clarified.  And yes, we agree and now mention the caveats on the drug and diet associations. Although these were not part of the present study, we do mention this in the discussion.

Round 2

Reviewer 4 Report

In discussion we added in thrst review:

It is described: 1/ Altered behavior, brain structure, and neurometabolites in a rat model of autism-specific maternal autoantibody exposure.Matthew R Bruce et al  Mol Psychiatry. 2023 Mar 27.doi: 10.1038/s41380-023-02020-3. 2/ Novel maternal autoantibodies in autism spectrum disorder: Implications for screening and diagnosis.Mazón-Cabrera R,et al. Front Neurosci. 2023 Feb 2;17:1067833. doi: 10.3389/fnins.2023.1067833. eCollection 2023. PMID: 36816132.

This information is not updated or it is not justified in case it is wrong

Author Response

IN the previous round, the reviewer queried:

-maternal antibody response might be related due to the known relationship between stress and immune function (Kiecolt-Glaser et al., 2007). To our knowledge, no studies have examined the relationship between these two factors. It is described: 1/ Altered behavior, brain structure, and neurometabolites in a rat model of autism-specific maternal autoantibody exposure.Matthew R Bruce et al  Mol Psychiatry. 2023 Mar 27.doi: 10.1038/s41380-023-02020-3. 2/ Novel maternal autoantibodies in autism spectrum disorder: Implications for screening and diagnosis.Mazón-Cabrera R,et al. Front Neurosci. 2023 Feb 2;17:1067833. doi: 10.3389/fnins.2023.1067833. eCollection 2023. PMID: 36816132.

We responded by stating that it was our hypothesis that these factors are related, and that the relationship is not yet known.

In THIS SECOND round the reviewer queried:

It is described: 1/ Altered behavior, brain structure, and neurometabolites in a rat model of autism-specific maternal autoantibody exposure.Matthew R Bruce et al  Mol Psychiatry. 2023 Mar 27.doi: 10.1038/s41380-023-02020-3. 2/ Novel maternal autoantibodies in autism spectrum disorder: Implications for screening and diagnosis.Mazón-Cabrera R,et al. Front Neurosci. 2023 Feb 2;17:1067833. doi: 10.3389/fnins.2023.1067833. eCollection 2023. PMID: 36816132.

This information is not updated or it is not justified in case it is wrong

Response:

These two papers discuss the important pathophysiology of maternal antibodies to fetal brain.  We have added these now to the discussion.  However, neither of these papers discuss the relationship between prenatal stress exposure and maternal antibodies to fetal brain, which is the novel aspect of the current work.
